# Pricing and Competition for Generative AI

**Rafid Mahmood**
NVIDIA & University of Ottawa
`rmahmood@nvidia.com`

## Abstract

Compared to classical machine learning (ML) models, generative models offer a new usage paradigm where (i) a single model can be used for many different tasks out-of-the-box; (ii) users interact with this model over a series of natural language prompts; and (iii) the model is ideally evaluated on binary user satisfaction with respect to model outputs. Given these characteristics, we explore the problem of how developers of new generative AI software can release and price their technology. We first develop a comparison of two different models for a specific task with respect to user cost-effectiveness. We then model the pricing problem of generative AI software as a game between two different companies who sequentially release their models before users choose their preferred model for each task. Here, the price optimization problem becomes piecewise continuous where the companies must choose a subset of the tasks on which to be cost-effective and forgo revenue for the remaining tasks. In particular, we reveal the value of market information by showing that a company who deploys later after knowing their competitor's price can always secure cost-effectiveness on at least one task, whereas the company who is the first-to-market must price their model in a way that incentivizes higher prices from the latecomer in order to gain revenue. Most importantly, we find that if the different tasks are sufficiently similar, the first-to-market model may become cost-ineffective on all tasks regardless of how this technology is priced.

## 1 Introduction

The recent explosion of generative artificial intelligence (AI) has introduced new machine learning (ML) frameworks for applications from chatbots to robotics [Wu et al., 2023, Nasiriany et al., 2024]. Whereas in classical ML, a user interacted with a single model designed for a specific predictive task (e.g., classification) via input data and output predictions, a single generative AI model can solve a variety of tasks for a user out-of-the-box [Brown et al., 2020]. Moreover, users interact with the generative model over a universal interface of natural language prompting [Arora et al., 2022].

The prompt-based paradigm has fostered two recent human-AI interaction trends. First, prompting facilitates such a wide distribution of tasks (i.e., user inputs and model outputs) that conventional metrics for evaluating models have become insufficient, leaving the most effective evaluation metric to be a binary score of whether the user is satisfied with the model output [Li et al., 2024, Chiang et al., 2024]. For example, Ziegler et al. [2024] empirically analyzed the GitHub Copilot software to reveal that the frequency of generated code approved by a user *is a better predictor of perceived [user] productivity than alternative measures.* Second, if a user does not receive a satisfactory output, they can try again in another prompting round by inputting to the model additional information [Castro et al., 2023]. For instance, the Anthropic HH and the Chatbot Arena datasets report on average 2.3 and 1.3 prompting rounds per conversation, respectively [Bai et al., 2022, Chiang et al., 2024].

In this work, we study the impact of these interaction characteristics on the pricing of generative AI technology. While classical ML products can be priced by analyzing the user demand for a model that can achieve a given performance metric on a specific task [Gurkan and de Véricourt, 2022, Mahmood

38th Conference on Neural Information Processing Systems (NeurIPS 2024).

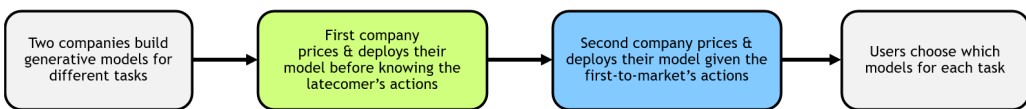

Figure 1: Overview of the competitive pricing problem for generative AI models.

et al., 2022], a generative AI model is priced per user prompt [1]. This set price determines the user cost for multiple different tasks and variable number of prompting rounds, e.g., the cost of using GPT-4 for math reasoning or code generation depends only on the per-token price, and the length and number of prompts. Thus, developers of a generative AI product must factor the demand for all potential use-case tasks of the technology when setting a price. This pricing problem becomes further challenging when considering the rapidly growing marketplace of competing generative AI models, since companies must also ensure that their products do not become unattractive to users as soon as a competitor develops a newer and better model.

We first characterize when, for a given task, a user will prefer one generative AI model versus another. We argue that users minimize their total cost, measured by the cost-per-prompt times the number of prompting rounds needed for the model to produce a satisfactory output; this leads to a comparison of price-performance competitiveness between AI models. We then study a game with two firms developing competing models used for a set of tasks. Both firms know each other's model's performance on the tasks. The first firm deploys their product and sets a price, followed by the second firm with their product and price. Finally, a user decides which models to use for each task. Both firms seek to maximize revenue, but the first firm acts without knowledge of their competitor's price. Figure 1 summarizes the problem setting and insights. Our key observations include:

1. **The pricing problem reduces to a piecewise optimization problem, where firms price their model to be competitive on a subset of the tasks while forgoing revenue from the others.** This subset can be determined by ranking the tasks on the competitive ratio between the two models for each task and selecting the most competitive tasks.

2. **A firm who deploys late always obtains revenue from at least one task by leveraging the available market information.** In contrast, the first-to-market must strategically set their prices to encourage the latecomer to set higher prices and focus on fewer tasks.

3. **Under certain conditions on model performance and user demand, the first-to-market may acquire zero revenue regardless of their price.** In these settings, the latecomer naturally maximizes their revenue by being competitive for all tasks. Thus, developers that are first should have a minimum model performance before deploying their product.

## 2 Related literature

**Evaluating ML models.** ML models are typically evaluated on generalization error for a task via out-of-sample test dataset benchmarks and competitions [Deng et al., 2009]. Generative AI and large language models (LLMs) are evaluated on a suite of benchmark tasks such as for coding [Chen et al., 2021], math [Cobbe et al., 2021], and problem solving [Hendrycks et al., 2021]. However, standardized benchmarks become uninformative over time as new models are trained to overfit to these metrics [Roelofs et al., 2019, Koch et al., 2021]. Recently, Chiang et al. [2024] introduced the Chatbot Arena for comparing LLMs head-to-head on human preference. Here, a user poses a real-world prompt which is input to two models. The user reviews both model outputs and can even continue multiple conversation rounds, before ranking which model generated a satisfactory answer first. Although difficult to measure, user satisfaction rate is increasingly viewed as the most informative metric as seen from generated code approvals on GitHub Copilot [Ziegler et al., 2024], or perceived aesthetic quality of text-to-image generation such as MidJourney and Playground [Li et al., 2024]. Our work combines user satisfaction with a user cost to construct a price-performance ratio for comparing different models.

---

[1]In practice, generative AI models are typically priced-per-token. In Appendix B, we show that all our results extend to the price-per-token setting with a minor change of variables. For a list of prices for current generative AI prices, see: `https://docsbot.ai/tools/gpt-openai-api-pricing-calculator`.

**Human-generative AI interaction.** Prompt-based interaction has increased the diversity of tasks where these models can be applied [Eloundou et al., 2023]. Castro et al. [2023] analyze how and when human users will use a generative AI model for a task versus performing it manually, as well as the characteristics of interacting over multiple prompt rounds. The quality of prompts is crucial to generating higher-quality answers [Liu et al., 2023, Binz and Schulz, 2023]. This has motivated studies on prompt techniques, such as chain-of-thought [Wei et al., 2022] and self-consistency [Wang et al., 2022]. In our work, we treat prompt quality as a random variable and to simplify the structural analysis, assume that users will interact with a generative AI model for as many prompting rounds as needed to get a satisfactory answer.

**Pricing and competition.** Duopolies of competitive products use game theoretic models such as the Bertrand (i.e., simultaneous pricing) and Stackelberg (i.e., sequential pricing) models of interaction [Gibbons, 1992]. Both deterministic and probabilistic demand can be used to study oligopolistic pricing of a single or multiple products Chintagunta and Rao [1996], Gallego and Hu [2014]. Specific structured demand frameworks allow for identifying market equilibria and failure settings where revenue is unobtainable [Federgruen and Hu, 2015, 2019]. Our work is most closely related to the Stackelberg literature by modeling a sequential game and exploring the conditions under exponential demand that make certain generative AI models unattractive [Hamilton and Slutsky, 1990].

**Pricing AI products.** AI technology can be priced at various levels, ranging from training data to model queries [Liu et al., 2021, Chen et al., 2019, Cong et al., 2022]. A core aspect of the pricing problem involves valuating the ML model based on performance [Xu et al., 2024]. ML products are further susceptible to an AI flywheel effect where the release and price of an AI product will affect the subsequent collection of new training data from users, leading to a dynamic pricing problem [Gurkan and de Véricourt, 2022, Chen and Xue, 2023]. More generally, novel technology products such as a new generative AI model with emergent use-cases may feature social-learning and dynamically growing market sizes [Feldman et al., 2019, Zhang et al., 2022]. The closest to our work is Gurkan and de Véricourt [2022] who explore pricing and contracting the development of a classical ML model under the AI flywheel. In contrast, our work explores competition between ML model developers when faced with a diverse set of potential downstream tasks for which the model can be used.

# 3 Main model

We first define the characteristics of the pricing problem. We then propose a model of user choice between two competing models from a price-performance perspective.

## 3.1 Problem setup

**Tasks.** We define a task as a set of independent problem instances where for each instance, a user queries a machine learning model with an input prompt and receives an output generated the model. For example, a programming task may have instances where a user inputs a commented function definition and the model must complete the code to perform the function Chen et al. [2021], Ziegler et al. [2024]. Task instances are evaluated by a user via a binary correctness score. For tasks where correctness is unambiguous (e.g., whether the program runs), the score is equivalent to accuracy, whereas for open-ended tasks (e.g., whether the output meets the stylistic preferences of the user), we treat correctness simply as whether the user is satisfied with the output [2]

**Generative AI model.** Given a set of $T$ different tasks, a generative model is an ML model that can be used to solve instances of any of the different tasks via prompts. We define this model as a tuple $(p, V_1, V_2, \ldots, V_T)$ where $p$ denotes the price for using the model, as measured in dollars-per-prompt (see Appendix B for the extension to pricing per-token), and for each $t \in [T]$, $V_t \in (0, 1)$ denotes the average score of the model over instances of each task. We assume that $V_t \to 1$ implies that the model can always generate a correct output for any task instance and $V_t \to 0$ implies that the model will always generate an incorrect output. Therefore, for any random instance of task $t$, $V_t$ can also be interpreted as a Bernoulli probability of the generative model producing a satisfactory output in a single attempt.

---

[2]For example in code completion, users prefer generated code that provides a good starting point for the users to improve on rather than code that is technically correct but confusing [Ziegler et al., 2024].

Users will not use the generative model if it's price is too high with respect to the user's valuation of the specific task. For any given task $t$, let $D_t(p) \in \mathbb{R}_+$ be the demand function, i.e., the number of users who will use a generative AI model for task $t$ as a function of the price $p$. Following standard assumptions, we assume that $D_t(p)$ is differentiable and non-increasing in the model price, as well as being known to the developer of the AI model [Gallego and Van Ryzin, 1994].

**Multi-round use.** Most ML benchmarks typically evaluate models on whether the models can generate the correct output under a single prompt round [Chen et al., 2021, Cobbe et al., 2021, Hendrycks et al., 2021]. However, in-the-wild users of generative models typically have interactive multi-round conversations where if the model generates an unsatisfactory solution after the first prompt, the user can provide feedback via their preferences or corrections in a second prompt round [Castro et al., 2023, Liao et al., 2024]. For example in code completion, if the model fails to account for an edge-case input to the function, the user can identify the edge-case and ask the model to account for it. To characterize this multi-round use, we extend the single prompt to a sequence of Bernoulli trials that continue until the user is satisfied with the model output. For simplicity of modeling, we make the following assumptions on user behavior.

**Assumption 1.** *The total number of prompting rounds $n_t(V_t)$ that a user will engage with the model: (i) has a finite mean; and (ii) is independent of the model price $p$ conditioned on the user knowing $V_t$.*

Assumption 1 implies that the price of a model only determines demand via whether the model is used at all, rather than how many times (i.e., prompting rounds) the model is used. Under this assumption, there are many choices for modeling the distribution of $n_t(V_t)$. We give three examples:

- *Geometric:* Because non-expert users tend to design uninformative prompts [Zamfirescu-Pereira et al., 2023], we may suppose the probability of success on any round does not depend on user input, and each round is an i.i.d. Bernoulli trial with probability $V_t$. Then, the total number of rounds is $n_t \sim \mathrm{Geom}(V_t)$, following a Geometric distribution.

- *Truncated Geometric:* Users may quit the model after a maximum $T_t$ rounds if it fails to generate a satisfactory response [Castro et al., 2023]. For instance, conversations in the Chatbot Arena dataset terminate after an average 1.3 rounds [Chiang et al., 2024]. Here, $\Pr\{n_t = n\} := (1 - V_t)^{n-1} V_t$ for $1 \leq n < T_t$ and $\Pr\{n_t = T_t\} := (1 - V_t)^{T_t - 1}$.

- *Prompt-dependent:* Suppose the success probability is prompt-dependent $V_t(x_i)$ where $x_i \sim \Pr\{x\}$ is the prompt on the $i$-th round. If users prompt until the model generates a satisfactory answer, we have $\Pr\{n_t = n\} := \mathbb{E}_{x_1, \cdots, x_n}[V_t(x_n) \prod_{i=1}^{n-1} (1 - V_t(x_i))]$.

Ultimately, the choice of characterizing $n_t$ depends on the information available to the generative AI model provider. With limited information, the Geometric assumption may be most practical, but given knowledge of user prompts, we may consider more sophisticated models. Our results all hold independent of the distribution as long as Assumption 1 is satisfied. For ease of notation and interpretability, we assume $n_t \sim \mathrm{Geom}(V_t)$ in the remainder of this work.

### 3.2 Modeling user preference between AI models

Given the above task and user behavior framework, we analyze the problem of a user who must choose between two competing generative models to solve instances of their tasks. Since a user may prefer different models for different tasks, we consider a single task and omit subscript $t$.

Consider two generative models: model A $(q, W)$ and model B $(p, V)$. Under Assumption 1, given a sufficient number of prompt rounds, both models can eventually solve every task instance. Thus, a rational user will seek to minimize the expected cost of completing a task instance, measured by the average price-per-prompt times the expected number of rounds required to complete the task instance. In this comparison, model B will incur a lower cost for the user if

$$p\mathbb{E}[n(V)] \leq q\mathbb{E}[n(W)] \implies \frac{p}{V} \leq \frac{q}{W}. \tag{1}$$

Otherwise, model A incurs lower user costs. We assume that model B is preferred in ties. Note that the implication follows from our Geometric assumption of $n(V)$.

Condition (1) states that for any task, a specific generative AI model is preferred if the price-performance ratio for this task (i.e., the cost of using this model over the model's performance) is

lower than any other competing generative model. For example, if model B is twice as likely to generate a user-satisfactory output as model A for a given input prompt, i.e., $V = 2W$, then the user will prefer model B as long the cost of prompting this model is not twice as high, i.e., $p \leq 2q$. Otherwise, the user will incur lower costs and still obtain their desired outputs by simply prompting the weaker model for twice as many rounds.

We note that (1) can also compare the use of a generative AI model versus manually performing the task [Castro et al., 2023]. For instance, if we treat model B as a human and a prompt round as a timed attempt at completing a task (e.g., coding the function within one hour), then $V$ is the probability of a human being able to perform the task in the single attempt and $p$ is the time-value of this labor.

Finally, this framework can also compare free-to-use generative AI models such as LLaMA [Touvron et al., 2023]. Although there may not be a given price-per-token for using these models, there is a fixed cost to set up the infrastructure and environment. Given an expected total number of task instances, this fixed cost can be approximated to an equivalent $p$.

## 4 Pricing generative AI models

We now develop a general framework under which a provider of a generative AI model can price their product. We first define the pricing problem as a game between two firms developing competing generative AI models. We then create tractable reformulations for these problems for both firms, representing pricing with and without considering competition.

**Pricing problem.** Consider a set of $T$ tasks. Let $(q, W_1, W_2, \ldots, W_T)$ and $(p, V_1, V_2, \ldots, V_T)$ be the generative models released by two competing firms, A and B, respectively. Firm A deploys their generative AI model (i.e., model A) first and sets the price $q$ for this product. Then, firm B deploys their competing model (i.e., model B) and sets their price $p$. After both firms deploy their models, a user with demand functions $D_t(\cdot)$ for each $t$ will decide which models to use as determined by (1). We evaluate the total revenue obtained by each firm, given by $R_A(q|p)$ and $R_B(p|q)$ respectively:

$$R_A(q|p) := q \sum_{t=1}^{T} D_t(q) \mathbb{1} \left\{ \frac{q}{W_t} < \frac{p}{V_t} \right\} \qquad R_B(p|q) := p \sum_{t=1}^{T} D_t(p) \mathbb{1} \left\{ \frac{p}{V_t} \leq \frac{q}{W_t} \right\}. \qquad (2)$$

Note that in practice, the user demand may depend on $p$ and $q$ simultaneously; for simplicity, we assume the demand for a specific model to be determined only after the user preference condition (1) is resolved. The results in this section easily generalize to more complex demand functions. The revenue functions are composed of the demand for each task times the price set by the firm [Van Ryzin and Talluri, 2005], summed over all tasks for which the firm's model is competitive according to the price-performance ratios. Each firm's objective is to maximize their revenue. However, because firm A is the first-to-market and they do not know the action that firm B will take; instead, firm A must optimize for the worst-case scenario as determined by firm B. On the other hand, firm B first observes the price set by firm A. Thus, the two firms set prices as follows:

$$q^* := \arg\max_{q \geq 0} \left\{ R_A(q|\hat{p}) \mid \hat{p} \in \arg\max R_B(p|q) \right\} \qquad p^* := \arg\max_{p \geq 0} \left\{ R_B(p|q^*) \right\}. \qquad (3)$$

We assume both firms know $V_t$ and $W_t$ for all $t \in [T]$. This is motivated by the availability of research papers and reported benchmark scores, and the predictability of model performance via power laws [Kaplan et al., 2020]. In practice, a firm may not know their competitor's performance, but they can forecast the state-of-the-art score in the short term.

### 4.1 Pricing in isolation

We first consider firm B's problem, who set their price after firm A has already released a competing model with price $q^*$. Firm B's problem depends on the competitive ratio $\kappa_t := \mathbb{E}[n(W_t)]/\mathbb{E}[n(V_t)]$ for task $t \in [T]$, i.e., the relative ratio of number of prompting rounds between the two firms, which under a Geometric distribution assumption, simplifies to $\kappa_t := V_t/W_t$. To maximize their revenue, firm B can rank the tasks in order of $\kappa_t$, select a subset of tasks in this order, and solve the optimization problem for each subset. This results in an overall piecewise optimization problem.

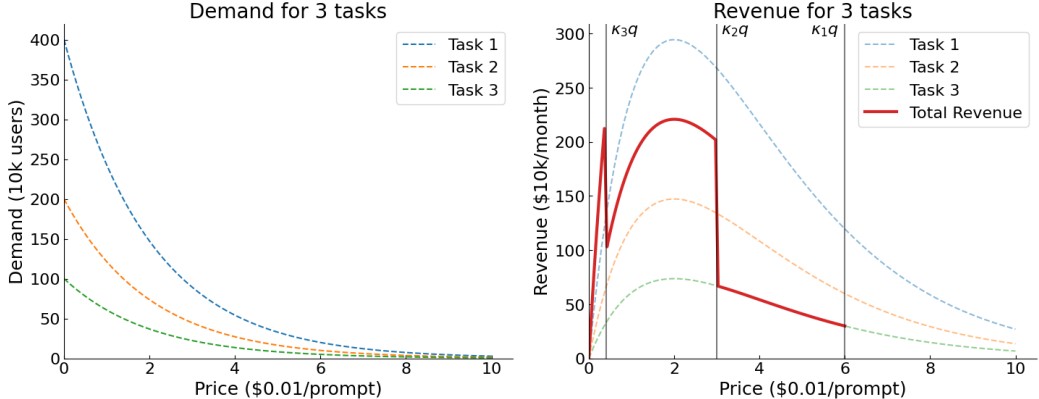

Figure 2: *(Left)* Three tasks with three different exponential demand functions $D_1(p) = 100e^{-0.5p}$, $D_2(p) = 200e^{-0.5p}$, $D_3(p) = 400e^{-0.5p}$. *(Right)* The corresponding revenue from each task along with the total revenue function for a firm $R_B(p)$. The vertical lines correspond to $\kappa_1 q$, $\kappa_2 q$, and $\kappa_3 q$, where $\kappa_1 > \kappa_2 > \kappa_3$. For $p < \kappa_3 q$, revenue is obtained from all three tasks, for $p \in (\kappa_3 q, \kappa_2 q]$, revenue is obtained from only the first two tasks, and for $p \in (\kappa_2 q, \kappa_1 q]$, revenue is only obtained from the first task. No revenue can be obtained if $p > \kappa_1 q$.

**Theorem 1.** *Consider the following ordering $\sigma : [T+1] \to [T+1]$ for which*

$$\kappa_{\sigma(1)} > \kappa_{\sigma(2)} > \cdots > \kappa_{\sigma(T)} > \kappa_{\sigma(T+1)} := 0. \tag{4}$$

*Then, firm B's pricing problem is equivalent to the piecewise optimization problem:*

$$\max_{t \in [T]} \max_p \left\{ p \sum_{s=1}^t D_{\sigma(s)}(p) \mid \kappa_{\sigma(t)} \geq \frac{p}{q} > \kappa_{\sigma(t+1)} \right\}. \tag{5}$$

Theorem 1 shows that a generative AI model should be priced by prioritizing a subset of the tasks and making the model price-performance competitive for those tasks only. Consequently, the firm ignores the remaining tasks for which the model has low competitive ratios $\kappa_t$, since they would need extremely low prices in order to satisfy (1) for these tasks. This strategy is due to the observation that for any task $\sigma(t)$, if a model satisfies (1), then the model will also satisfy the condition for all $\sigma(t')$ for $t' \leq t$. Furthermore, Theorem holds without loss of generality with respect to the strict inequalities on (4), since tasks with the same competitive ratios can be grouped together by summing the constituent demand functions. Finally, Theorem 1 holds for arbitrary demand functions. In practice, the demand functions are known and typically have a parametric structure. Figure 2 gives an example using three tasks with demand that decays exponentially with price.

Theorem 1 reveals two key implications on how a firm can price a generative mode when given competitor information. First, pricing reduces to solving $T$ optimization problems, where each of these problems are of a single variable with a differentiable objective and boundary constraints. Thus, the inner problems can be solved via gradient descent. Second, as long as firm B sets a price $p \leq \kappa_{\sigma(1)}$, they will obtain some non-zero revenue, i.e., problem (5) always has a feasible solution. This advantage is due to the fact that firm B sets their price given a fixed $q$.

### 4.2 Pricing when accounting for competition

We next consider firm A's problem of setting a problem while assuming that firm B will act optimally next. This pricing problem is a bi-level optimization problem, but it can be solved by ranking the tasks according to the competitive ratios for firm A and prioritizing a subset of these tasks.

**Theorem 2.** *Firm A's pricing problem is equivalent to the piecewise bi-level optimization problem:*

$$\max_{t \in [T-1]} \max_{q \geq 0} \quad q \sum_{s=t+1}^{T} D_{\sigma(s)}(q)$$

$$\text{s.t.} \quad \max_{p} \left\{ p \sum_{s=1}^{t} D_{\sigma(s)}(p) \,\middle|\, \kappa_{\sigma(t)} \geq \frac{p}{q} > \kappa_{\sigma(t+1)} \right\} \tag{6}$$

$$> \max_{p'} \left\{ p \sum_{s=1}^{t'} D_{\sigma(s)}(p') \,\middle|\, \kappa_{\sigma(t')} \geq \frac{p'}{q} > \kappa_{\sigma(t'+1)} \right\} \quad \forall t' \neq t$$

Theorem 2 relies on the observation that the reverse order of $\sigma(\cdot)$ in (4) cn rank the most to least competitive tasks for firm A. For any $t$, if $q < p\kappa_{\sigma(t)}^{-1}$, then model A is price-performance competitive for all tasks $\sigma(t), \cdots, \sigma(T)$ and firm A will acquire revenue from all these tasks.

Firm A may not always be able to obtain revenue. Problem (6) is infeasible if for every $t \in [T-1]$, there is no $q \geq 0$ that satisfies the bi-level constraint. This infeasibility implies for any $q \geq 0$, firm B always maximizes their revenue by setting a low price $p \leq \kappa_{\sigma(T)}q$. Thus, the key motivation of firm A is that the firm benefits only when their competitor is incentivized to set high prices.

## 5 Structural analysis under exponential demand

Demand is typically modeled via structured parametric functions [Van Ryzin and Talluri, 2005]. In this section, we specialize the pricing problem to the standard choice of exponentially decaying parametric demand to extend the previous general results. See Figure 2 (Left) for an example.

**Assumption 2.** *For each task $t$, the demand function decays exponentially in price $D_t(p) := a_t \exp(-bp)$, where $a_t > 0$ is the zero-price base demand and $b > 0$ is the price-sensitivity of users. Furthermore, all tasks have the same price-sensitivty.*

Under the exponential demand model, the demand for each task $t$ is equal to $a_t$ when $p = 0$, and decays at a rate $-b$. We assume that the decay rate is the same for each task; this is motivated by the practical consideration that the different tasks should have relatively similar 'user value' to have the same price. If one task is uniquely price-sensitive to users, the firm may instead develop a finetuned model for that task or propose incentives such as task-specific discounts to better optimize revenue.

Below, we revisit both firm B's and firm A's problems under this demand model to derive globally optimal solutions and structural insights on the market dynamics.

### 5.1 Pricing in isolation under exponential demand

Under Assumption 2, firm B's problem (5) now simplifies to the maximum of up to $T$ possible values that are the globally optimal solution to each of the individual inner optimization problems.

**Theorem 3.** *Suppose Assumption 2 holds. For any $t$, let $\bar{a}_{\sigma(t)} := \sum_{s=1}^{t} a_{\sigma(t)}$. Without loss of generality, let $t^* \in [T]$ be the task index that satisfies $\kappa_{\sigma(t^*)}q \geq 1/b > \kappa_{\sigma(t^*+1)}q$. Then, firm B's pricing problem is equivalent to the following maximum value:*

$$\max \left( \max_{t > t^*} \left\{ \kappa_{\sigma(t)}q\bar{a}_{\sigma(t)}e^{-b\kappa_{\sigma(t)}q} \right\} , \frac{1}{b}\bar{a}_{\sigma(t^*)}e^{-1} \right) \tag{7}$$

Theorem 3 states that problem (5) can be solved by by solving each of the inner problems starting from the lowest price range up until we arrive at the zero-gradient solution of the revenue function, i.e., $1/b$. Furthermore, for each of the price ranges before this point, the optimal solution is the upper price boundary. Note that if $1/b > \kappa_{\sigma(1)}q$, i.e., there does not exist any $t^*$ satisfying the condition, then we take the maximum of all $T$ problems. Figure 2 (Right) visualizes the revenue function $R_B(p|q)$ for $T = 3$ tasks with exponential demand. Finally, this structure also reveals that the optimal price is bounded from both above and below via these optima.

**Corollary 1.** *The optimal price for firm B is bounded $1/b \geq p^* \geq \kappa_{\sigma(1)}q$.*

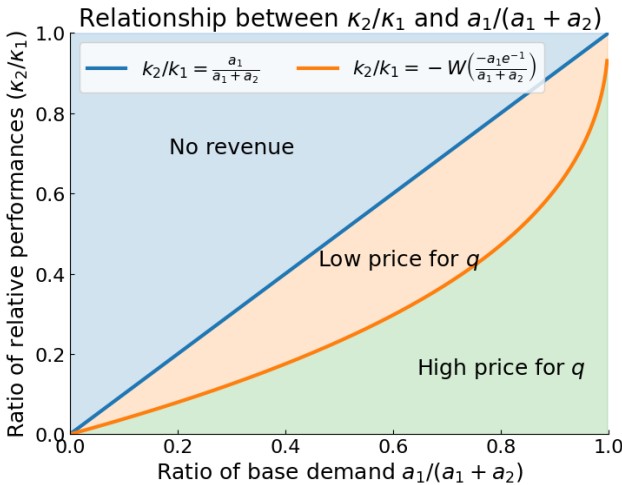

Figure 3: The relationship between $\kappa_2/\kappa_1$ and $a_1/(a_1 + a_2)$ for firm A. In the blue region, firm B will always set a price that is competitive on both tasks and firm A will acquire zero revenue. In the orange region, the maximum price that firm A can set is upper bounded (see problem (11)). In the green region, the maximum price that firm A can set has a higher upper bound (see problem (10)).

## 5.2 Pricing when accounting for competition under exponential demand

We now revisit firm A's pricing problem, which is a bi-level problem with multiple inner optimization constraints. To obtain structural insights on the challenges of pricing under competition, we explore a special case with $T = 2$ tasks. Without loss of generality, we assume $\kappa_1 > \kappa_2$, i.e., $\sigma(t) = t$.

When there are only two tasks, firm B will either set their price to be competitive for the first task only (i.e., the task with the higher competitive ratio) or for both tasks. In the latter case, model A would be unattractive for both tasks and firm A would obtain zero revenue. Therefore, firm A should set their price in such a way that they maximize revenue on the second task, while ensuring that firm B is motivated to be competitive only for the first task. Thus, problem (6) simplifies to

$$
\begin{aligned}
\max_{q} \quad & qD_2(q) \\
\text{s.t.} \quad & \max_{p} \left\{ pD_1(p) \mid \kappa_1 q \geq p > \kappa_2 q \right\} > \max_{p} \left\{ p\left(D_1(p) + D_2(p)\right) \mid \kappa_2 q \geq p > 0 \right\}
\end{aligned}
\tag{8}
$$

If firm A sets a price $q$ that is infeasible for this problem, they will get zero revenue. Furthermore, this problem reduces to two single-level optimization problems with only bounding constraints.

**Theorem 4.** *Suppose that $T = 2$, that Assumption 2 holds, and without loss of generality, assume $\sigma(t) = t$. For $z \in \mathbb{R}$, let $\mathcal{W}(z)$ be the Lambert $\mathcal{W}$ function defined only over $z > -e^{-1}$. If*

$$
\frac{\kappa_2}{\kappa_1} \leq -\mathcal{W}\left(-\frac{a_1 e^{-1}}{a_1 + a_2}\right)
\tag{9}
$$

*then, the firm A's pricing problem is equivalent to*

$$
\max\left\{ qa_2 e^{-bq} \; \middle| \; -\frac{1}{b\kappa_2}\mathcal{W}\left(-\frac{a_1 e^{-1}}{a_1 + a_2}\right) \geq q > 0 \right\}.
\tag{10}
$$

*Otherwise, firm A's problem is equivalent to*

$$
\max\left\{ qa_2 e^{-bq} \; \middle| \; \frac{1}{b(\kappa_1 - \kappa_2)}\left(\log\frac{\kappa_1}{\kappa_2} + \log\frac{a_1}{a_1 + a_2}\right) \geq q > 0 \right\}.
\tag{11}
$$

Recall from Theorem 3, the second-level problems in (8) can be solved analytically by checking the boundary points and zero-gradient solution. Theorem 4 uses this property to map problem (8) to two sub-problems based on whether condition (9) holds.

Determining which problem to solve to obtain the optimal price depends on a relationship between two constants $\kappa_2/\kappa_1$ and $a_1/(a_1 + a_2)$. Here, $\kappa_2/\kappa_1$ is the relative competitive ratio with respect to

the two tasks for firm B, where $\kappa_1 > \kappa_2 > 0$. Thus, $\kappa_2/\kappa_1 \to 1$ suggests that the relative performance differences between model A and model B are similar for both tasks, whereas $\kappa_2/\kappa_1 \to 0$ suggests that relative to model A, model B's performance is much worse on the second task than for the first task. The second parameter $a_1/(a_1 + a_2)$ is the fraction of the total demand that is occupied by the first task. If this is close to 1, then the first task has significantly higher demand than the second, but if it is close to 0, then the first task has significantly lower demand than the second.

We now discuss the structural insights obtained from Theorem 4 (see Figure 3 for a visualization). First, note that when condition (9) is satisfied, firm A is able to set higher prices than it could if the condition were not satisfied. In the latter case, the optimal price that firm A can set must be upper bounded by the constraint in problem (11), which in this scenario, is less than the upper bound in problem (10). Intuitively, this condition partitions firm A's pricing problem into two regimes: a high-price and low-price regime. Furthermore, the high-price regime is only attainable if the relative performance difference between the two tasks is greater than the Lambert $\mathcal{W}$ function of the fraction of total demand occupied by the first task.

Second, if firm B's relative performance difference between the two tasks is larger than the fraction of demand occupied by the first task, then firm A's pricing problem is infeasible. That is, no matter what price that firm A sets, firm B is always incentivized to set a price that ensures users will prefer model B and consequently, leave no revenue for firm A.

**Proposition 1.** *If $\kappa_2/\kappa_1 > a_1/(a_1 + a_2)$, then for any price that firm A sets, firm B will always set a price that allows them to be price-performance competitive for both tasks.*

Intuitively, if the relative performance difference is high, then model B's performance relative to model A is approximately the same for both tasks. Consequently, the range $(\kappa_2 q, \kappa_1 q]$ is small, meaning that a small perturbation from this price would not significantly decrease firm B's revenue from the first task, while permitting the firm to obtain revenue from the second task as well. Note that model B does not necessarily have to be 'better' than model A, only that the performance on the tasks must be similar. Finally, this ratio $\kappa_2/\kappa_1$ only needs to be larger than the fraction of total demand that is occupied by the first task. Consequently, even if the ratio is small, firm B can still be incentivized to set a competitive price for both tasks if the potential revenue that can be obtained from the second task is high.

Theorem 4 and Proposition 1 sets a guideline for when a firm should deploy their generative AI model. Consider a company that has developed a model for a new application area with no competitors. This company knows that upon releasing their model to the public, competitors will release their own models. If the first-to-market company believes that the use-cases for this model are sufficiently similar to each other with respect to model performance, but are differentiated with respect to user demand, then they must ensure that the model performs exceedingly well on at least one specific use-case that their competitors cannot match. That is, the company must differentiate their product [Schmalensee, 1982], or competitors can outprice the initial company with similar products.

Although firm A must price accounting for firm B's actions, if model A is sufficiently differentiated from model B, then firm A can acquire their maximum possible revenue. Recall that the zero-gradient optimal price for problem (8) is $q^* = 1/b$. Furthermore recall that if condition (9) is met, then firm A can set higher prices for their model. We show below that if condition (9) is satisfied and if $\kappa_2$ is sufficiently low with respect to the demand ratio, then the optimal price is feasible.

**Proposition 2.** *If $\kappa_2 \le -\mathcal{W}(a_1 e^{-1}/(a_1 + a_2)) \min(\kappa_1, 1)$, then firm A maximizes their revenue by setting $q^* = 1/b$.*

# 6    Conclusion

We study how a company developing a generative AI model can set the price for this technology. Generative AI models are a fundamentally different technology product compared to classical AI models due to two factors. First, a single generative AI model is performant on multiple distinct use-cases that each invite individual respective user demands. Second, generative AI models offer interactivity via prompting, thereby encouraging 'geometric' user interaction where users can repeatedly prompt the model until it generates a satisfactory answer, compared to classical non-interactive AI models that invite one-shot 'Bernoulli' interaction. These features combined with a singular unit price per-prompt for model use warrant new revenue maximization frameworks for this technology.

We find that for generative AI models, the pricing problem reduces to ranking the different tasks in order of the relative performance of the model versus a competing alternative. In isolation, a company can then always obtain non-zero revenue by setting a price to be competitive for at least one downstream task. However, when considering the competition from alternative models that may be released after the price is set, the company faces strict upper bounds on the prices they can set based on the performance of the company's model relative to the latecomers. In particular, if the relative difference in performances on the tasks is sufficiently small, then a competitor will always be able to outprice the company on all tasks. This result reveals that an outsized performance improvement on at least one of the downstream applications of a generative AI model is essential to maximizing revenue.

**Limitations.** We explore a theoretical market problem where firms know user demand and the performance of competing AI models. In practice, these would be estimated with some noise. Furthermore, we study a static marketplace where each firm sets their price once. In contrast, AI technologies feature a flywheel where a low price and an early release of a product can allow for a firm to collect more training data and improve their model downstream [Gurkan and de Véricourt, 2022]. In particular, the opportunity to acquire high-quality data can significantly improve model performance, thereby motivating the first mover position. Finally, we do not include the cost of developing the generative AI models themselves, but assumes that the fims have already decided to build these models. The market dynamics can change when considering how large of a model to build, how much resources (e.g., compute hours) to spend, or even whether to build a generative AI model. We see these scenarios as important future extensions.

**Societal impact.** Pricing of generative AI can significantly affect the democratization of generative AI technology. This paper explores the conditions that incentivize firms to develop or deploy this technology, e.g., when firms can obtain revenue. Setting appropriate prices for these models can allow for this technology to be more easily accessible to a wider set of users.

## Acknowledgments and Disclosure of Funding

The author thanks the NeurIPS 2024 editorial chairs and anonymous referees for providing valuable feedback that significantly improved this paper. The author also thanks David Acuna, Sanja Fidler, Marc Law, and James Lucas for providing insightful discussion and valuable suggestions on early versions of this paper.

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

# A   Proofs

*Proof of Theorem 1.*  Consider the ordering $\sigma$ defined by (4) and note that for any $t$, if we constrain $p$ to satisfy

$$\kappa_{\sigma(t)} \geq \frac{p}{q} > \kappa_{\sigma(t+1)},$$

then all of the tasks $\sigma(1), \sigma(2), \ldots, \sigma(t)$ become competitive for firm B's generative model compared to firm A's model, i.e., by satisfying (1), whereas tasks $\sigma(t+1), \ldots, \sigma(T)$ are competitive for firm A's model instead. As a result, for prices constrained in this range, the resulting problem becomes $\max_p p \sum_{s=1}^{t} D_{\sigma(s)}(p)$. Therefore, to solve the overall pricing problem, we only need to solve this resulting constrained problem for each value of $t$. □

*Proof of Theorem 2.*  From Theorem 1, for any value of $q$, there exists a $t \in [T]$ such that firm B will prioritize revenue from tasks $\sigma(1), \sigma(2), \cdots, \sigma(t)$. This value of $t$ will satisfy

$$\max_p \left\{ p \sum_{s=1}^{t} D_{\sigma(s)}(p) \;\middle|\; \kappa_{\sigma(t)}q \geq p > \kappa_{\sigma(t+1)}q \right\} > \max_{p'} \left\{ p' \sum_{s=1}^{t'} D_{\sigma(s)}(p') \;\middle|\; \kappa_{\sigma(t')}q \geq p' > \kappa_{\sigma(t'+1)}q \right\}$$

for all $t' \neq t$. Furthermore, given this $t$, firm A will only obtain revenue from tasks $\sigma(t+1), \sigma(t+2), \cdots, \sigma(T)$, thereby revealing the objective function. Finally, note that if $t = T$, then firm A will not acquire any revenue since firm B will be price-performance competitive on all tasks. Therefore, firm A's optimization problem is to simultaneously solve for $q$ and for $t \in [T-1]$. □

*Proof of Theorem 3.*  First, note that under an exponential demand function the revenue function for each piece $t$ of problem (5) is $p\bar{a}_{\sigma(t)} \exp(-bp)$, and therefore has a zero-gradient point at $p^* = 1/b$. Thus, when solving problem (5), we can solve the inner problem by breaking into 3 cases based on which $t$ to consider.

**For $t^*$ satisfying the condition.**  Here, the zero-gradient price $p^*$ is a feasible solution to the inner pricing problem, meaning that this must be the optimal price. Substituting this into the revenue function yields $\bar{a}_{\sigma(t^*)} \exp(-1)/b$.

**For $t > t^*$.**  Note that for any $t > t^*$, the price is constrained to be less than $\kappa_{\sigma(t^*+1)} < 1/b$. In this regime, the revenue function $p\bar{a}_{\sigma(t)} \exp(-bp)$ is monotonically increasing, meaning that the optimal price will be the maximum possible value, i.e., $p = q\kappa_{\sigma(t)}$. Substituting this into the revenue function yields $\kappa_{\sigma(t)}q\bar{a}_{\sigma(t)} \exp\left(-b\kappa_{\sigma(t)}q\right)$.

**For $t < t^*$.**  Here, the set of feasible prices is strictly greater than $\kappa_{\sigma(t^*)} > 1/b$. In this regime, the revenue function is monotonically decreasing, meaning that the optimal price will be the minimum value, i.e., $p \to q\kappa_{\sigma(t+1)}$, approaching from above. However, for any such setting, decreasing $p$ to equal the infimum would make firm B price-performance competitive for the $t+1$-th task as well and thereby yield a higher return. Therefore, the optimal solution for any $t < t^*$ can always be upper bounded by the optimal solution for $t + 1$, leading up to $t^*$. □

*Proof of Corollary 1.*  The upper bound follows from the proof of Theorem 3, as we show that the optimal revenue cannot be achieved without satisfying task $\sigma(t^*)$. The lower bound follows from observing the optimal solution for $t = T$. □

*Proof of Theorem 4.*  Let $p_1 := \arg\max\{pD_1(p) \mid \kappa_1 q \geq p > \kappa_2 q\}$. Furthermore, let $p_2 := \arg\max\{p(D_1(p) + D_2(p)) \mid \kappa_2 q \geq p > 0\}$. Although $p_1$ and $p_2$ depend on $q$, we omit this dependency to simplify the notation. We prove our theorem by exploring three regimes for $q$: $(0, 1/(b\kappa_1)]$, $(1/(b\kappa_1), 1/(b\kappa_2)]$, and $(1/(b\kappa_2), \infty)$. We then show that whether condition (9), the problems for each regime simplify further into problems (10) and (11).

**For $q \in (1/(b\kappa_2), \infty)$.**  For any $q > 1/(b\kappa_2)$, we have $\kappa_2 q \geq 1/b > 0$. From Theorem 3, $p_2 = 1/b$ is the optimal price. Furthermore, since $\kappa_1 q \geq p_1 > \kappa_2 q$, we have $p_1 > 1/b$. From Corollary 1, $p_1$ is priced too high and cannot achieve a higher revenue than $p_2$. Therefore in this regime, firm B will set a price low enough that their model is competitive for both tasks, and consequently firm A will achieve zero revenue.

**For** $q \in (1/(b\kappa_1), 1/(b\kappa_2)]$. From Theorem 3, $p_1 = 1/b$ and $p_2 = \kappa_2 q$ are the optimal prices that firm B can set for their two sub-problems. For firm A to achieve any revenue, the constraint in problem (8) becomes

$$\frac{1}{b}a_1 \exp(-1) > \kappa_2 q(a_1 + a_2)\exp(-b\kappa_2 q) \implies -b\kappa_2 q \exp(-b\kappa_2 q) > -\left(\frac{a_1 \exp(-1)}{a_1 + a_2}\right)$$

$$\implies -b\kappa_2 q > \mathcal{W}\left(-\frac{a_1 \exp(-1)}{a_1 + a_2}\right)$$

$$\implies q < -\frac{1}{b\kappa_2}\mathcal{W}\left(-\frac{a_1 \exp(-1)}{a_1 + a_2}\right) \leq \frac{1}{b\kappa_2}$$

Above, the first line follows from rearranging the terms and the second line follows from applying the definition of the Lambert $\mathcal{W}$ function, i.e., $\mathcal{W}(z) = y \Leftrightarrow y\exp(y) = z$. The third line follows again from rearranging the terms. Finally, note that the Lambert $\mathcal{W}$ function is bounded in $[-1, 0]$, meaning that this constraint dominates the original upper bound. Therefore, for $q$ in this regime, we can solve the problem

$$\max\left\{qa_2 \exp(-bq) \;\middle|\; -\frac{1}{b\kappa_2}\mathcal{W}\left(-\frac{a_1 \exp(-1)}{a_1 + a_2}\right) \geq q \geq \frac{1}{b\kappa_1}\right\} \qquad (12)$$

**For** $q \in (0, 1/(b\kappa_1)]$. From Theorem 3, $p_1 = \kappa_1 q$ and $p_2 = \kappa_2 q$ are the optimal prices that firm B can set when $q \leq 1/(b\kappa_1)$. Here, the constraint that ensures firm B will only prioritize the first task reduces to

$$q\kappa_1 a_1 \exp(-b\kappa_1 q) > q\kappa_2(a_1 + a_2)\exp(-b\kappa_2 q) \implies \frac{\kappa_1}{\kappa_2}\left(\frac{a_1}{a_1 + a_2}\right) > \exp(bq(\kappa_1 - \kappa_2)q)$$

$$\implies b(\kappa_1 - \kappa_2)q < \log\frac{\kappa_1}{\kappa_2} + \log\frac{a_1}{a_1 + a_2}$$

$$\implies q < \frac{\log\frac{\kappa_1}{\kappa_2} + \log\frac{a_1}{a_1 + a_2}}{b(\kappa_1 - \kappa_2)}.$$

Above, the first line follows from rearranging the terms and the second line follows from taking the log on both sides. The third line follows from rearranging the terms. Therefore, for $q$ in this regime, we can solve the problem

$$\max\left\{qa_2 \exp(-bq) \;\middle|\; \min\left(\frac{1}{b\kappa_1}, \frac{1}{b(\kappa_1 - \kappa_2)}\left(\log\frac{\kappa_1}{\kappa_2} + \log\frac{a_1}{a_1 + a_2}\right)\right) \geq q > 0\right\}. \qquad (13)$$

We now show that when condition (9) is satisfied, problems (12) and (13) join to have one continuous feasible set, whereas when the condition is not satisfied, problem (12) is infeasible and the minimum disappears.

First, to see that when the condition is satisfied, the two problems have one feasible set, we must prove

$$\frac{1}{b(\kappa_1 - \kappa_2)}\left(\log\frac{\kappa_1}{\kappa_2} + \log\frac{a_1}{a_1 + a_2}\right) \geq \frac{1}{b\kappa_1}. \qquad (14)$$

Specifically,

$$-\frac{\kappa_2}{\kappa_1} \geq \mathcal{W}\left(-\frac{a_1 \exp(-1)}{a_1 + a_2}\right) \implies \frac{\kappa_2}{\kappa_1}\exp\left(-\frac{\kappa_2}{\kappa_1}\right) \leq \frac{a_1 \exp(-1)}{a_1 + a_2}$$

$$\implies \exp\left(1 - \frac{\kappa_2}{\kappa_1}\right) \leq \frac{\kappa_1}{\kappa_2}\left(\frac{a_1}{a_1 + a_2}\right)$$

$$\implies 1 - \frac{\kappa_2}{\kappa_1} \leq \log\frac{\kappa_1}{\kappa_2} + \log\frac{a_1}{a_1 + a_2}$$

$$\implies \frac{\kappa_1 - \kappa_2}{\kappa_1} \leq \log\frac{\kappa_1}{\kappa_2} + \log\frac{a_1}{a_1 + a_2}$$

$$\implies \frac{1}{b\kappa_1} \leq \frac{1}{b(\kappa_1 - \kappa_2)}\left(\log\frac{\kappa_1}{\kappa_2} + \log\frac{a_1}{a_1 + a_2}\right).$$

Above, the first line follows from applying the Lambert $\mathcal{W}$ function. The second line follows from rearranging the terms and the third line follows from taking the log on both sides. The fourth and fifth lines follow from rearranging the terms and multiplying both sides by $1/b$. Note that when this condition is satisfied, the lower bound for problem (12) and the upper bound for problem (13) are equivalent, meaning that we can merge the two disjunctive regions. This results in obtaining problem (10).

Next, we first note that when the condition is not satisfied, the inequality in (14) is reversed and the minimum on the left-hand-side of the constraint in (13) can be removed. We then show that problem (12) becomes infeasible. Specifically,

$$\frac{\kappa_2}{\kappa_1} > -\mathcal{W}\left(-\frac{a_1 \exp(-1)}{a_1 + a_2}\right) \;\Rightarrow\; \frac{1}{\kappa_1} > -\frac{1}{\kappa_2}\mathcal{W}\left(-\frac{a_1 \exp(-1)}{a_1 + a_2}\right)$$
$$\Rightarrow\; \frac{1}{b\kappa_1} > -\frac{1}{b\kappa_2}\mathcal{W}\left(-\frac{a_1 \exp(-1)}{a_1 + a_2}\right)$$

The second line follows from multiplying both sides by $1/b$. This condition means that problem (12) does not have a feasible region, meaning that the optimal price can only be obtained by solving problem (11). □

*Proof of Proposition 1.* The proof of the follows from observing that when this condition is satisfied, the intermediate problems from the proof of Theorem 4, i.e., problems (12) and (13) both become infeasible. If those problems become infeasible, the nominal problems (10) and (11) also must be infeasible. First,

$$\frac{\kappa_1}{\kappa_2} < \frac{a_1 + a_2}{a_1} \;\Rightarrow\; \log \frac{\kappa_1}{\kappa_2} < \log \frac{a_1 + a_2}{a_1}$$
$$\Rightarrow\; \log \frac{\kappa_1}{\kappa_2} + \log \frac{a_1}{a_1 + a_2} < 0$$

The first line follows from taking the log and the second line follows from rearranging the terms and simplifying. When this condition is satisfied, problem (13) becomes infeasible.

Next, observe that for any $z \in [0,1]$, we have $z \geq -\mathcal{W}(z \exp(-1))$. Because $a_1/(a_1 + a_2) \in [0,1]$, for any value of $a_1$ and $a_2$, we have

$$\frac{a_1}{a_1 + a_2} \geq -\mathcal{W}\left(\frac{a_1 \exp(-1)}{a_1 + a_2}\right) \;\Rightarrow\; \frac{\kappa_2}{\kappa_1} > -\mathcal{W}\left(\frac{a_1 \exp(-1)}{a_1 + a_2}\right)$$
$$\Rightarrow\; \frac{1}{b\kappa_1} > -\frac{1}{b\kappa_2}\mathcal{W}\left(\frac{a_1 \exp(-1)}{a_1 + a_2}\right)$$

Thus, problem (12) is also infeasible. □

*Proof of Proposition 2.* We show that if this condition holds, then problem (10) is the active problem-to-solve and that the zero-gradient solution $1/b$ is a feasible solution. First, note that the condition implies that condition (9) holds, meaning that problem (10) is the problem-to-solve. We then note that condition implies

$$1 \leq -\frac{1}{\kappa_2}\mathcal{W}\left(-\frac{a_1 \exp(-1)}{a_1 + a_2}\right) \min\left(\frac{V_1}{W_1}, 1\right) \leq -\frac{1}{\kappa_2}\mathcal{W}\left(-\frac{a_1 \exp(-1)}{a_1 + a_2}\right)$$

Multiplying both sides by $1/b$ completes the proof. □

## B  Extensions from per-prompt to per-token pricing

In this paper, we assume that the generative AI models can be used at a fixed price per-prompt. In practice, generative AI models are priced either per-token or priced via a subscription mechanism where users pay a fixed cost for (potentially unlimited) use. Furthermore, different tasks may feature statistically different numbers of tokens either via longer (or shorter) prompts, or longer (or shorter) model outputs. Consequently under per-token pricing, different tasks will have different costs on

average. The price-per-prompt framework of this paper naturally extends to pricing per-token with a small change of variables that preserves all fundamental results.

Suppose there are two firms, A and B, developing generative AI models with fixed price per-tokens $q_0$ and $p_0$, respectively. We assume that the price is the same for both input and output tokens. For each task $t$ and model, there is now a distribution of the number of tokens $\phi_t$ and $\theta_t$, that the user sends and receives through the respective model in a given prompt. Note that the corresponding price-per-prompt $p_t$ and $q_t$ are now random variables $p_t = p_0\theta_t$ and $q_t = q_0\phi_t$, that depend on the specific task. For any given task $t$, a user will prefer model B if

$$p_0\mathbb{E}[\theta_t n(V_t)] \leq q_0\mathbb{E}[\phi_t n(W_t)]$$

where the expectation is taken over the randomness in the number of prompting rounds times the number of tokens per prompting round. The corresponding revenue functions for the two firms are

$$R_A(q_0|p_0) := q_0 \sum_{t=1}^{T} D_t(q_0)\mathbb{1}\left\{q_0\mathbb{E}[\phi_t n(W_t)] \leq p_0\mathbb{E}[\theta_t n(V_t)]\right\}$$

$$R_B(p_0|q_0) := p_0 \sum_{t=1}^{T} D_t(p_0)\mathbb{1}\left\{p_0\mathbb{E}[\theta_t n(V_t)] \leq q_0\mathbb{E}[\phi_t n(W_t)]\right\}.$$

To solve this problem, note that the inherent structure is equivalent to equation (3). Let $\kappa_t := \mathbb{E}[\phi_t n(W_t)]/\mathbb{E}[\theta_t n(V_t)]$ be the corresponding competitive ratio of the two models and note that the set of tasks can be ranked according to this re-defined competitive ratio. Then, Theorem 1 follows using the same steps and the re-defined $\kappa_t$. Moreover, all subsequent results carry over from this result.

