# OpenReview forum: "Pricing and Competition for Generative AI"
_NeurIPS.cc/2024/Conference — NeurIPS 2024 poster_

### Official Review · Reviewer_CY6H · 2024-07-12

**Soundness:** 3
**Presentation:** 3
**Contribution:** 2
**Rating:** 5
**Confidence:** 3

**Summary:**

The paper presents a theoretical analysis of pricing strategies for generative AI models in a competitive market. It investigates how companies can optimally set prices for AI models that are used across various tasks, considering the impact of competition from other firms. The study uses a game-theoretical approach to model the interactions between firms, exploring the implications of different pricing strategies on market dynamics and revenue optimization.

**Strengths:**

1.	The paper tackles an emerging and relevant problem in the field of AI economics, particularly the pricing of generative AI in a competitive landscape. The study is timely.
2.	The formulation is clear and the presentation is good. The paper in general is very easy to follow.
3.	The paper offers insightful conclusions about the strategic behaviors of firms in a duopoly, especially the advantages of being a second mover in the market.

**Weaknesses:**

1.	From the perspective of a stylized model, the work is solid and interesting. But I am not sure about the practice relevant. I am not sure whether the real world LLM company is thinking their pricing strategy in this way. I won’t push along this line, and would like leave the decision on this point to more senior reviewers and ACs.
2.	The main technical contribution to me is the modeling the GenAI’s prompt-based service. The analysis seems very standard.
3.	I am not fully convinced that in the current GenAI market, the companies are following the dynamics of model A and model B in the paper. All the companies should be keeping updated their accuracy and be almost homogenous in terms of pricing.

**Questions:**

See previous comments.

**Limitations:**

See previous comments.

---

> ### Author Rebuttal · Authors · 2024-08-04
>
> Thank you for your positive comments on the relevance and timeliness of our work, clarity of our formulation and writing, as well as the key insights drawn from our work.
>
> ---
>
> > W1) From the perspective of a stylized model, the work is solid and interesting. But I am not sure about the practice relevant. I am not sure whether the real world LLM company is thinking their pricing strategy in this way. I won’t push along this line, and would like leave the decision on this point to more senior reviewers and ACs.
>
>
> Thank you for the positive comments. Although we study the effects of competition, model performance, and user demand, we agree that the real-world pricing will likely incorporate additional factors (e.g., marketing, model development).
>
> Rather than algorithms for prescribing explicit prices, our goal is to reveal insights on market dynamics drawn from the unique traits of generative AI technology. Specifically, we show the importance of knowing competitor price and performance information on revenue. For example, Proposition 1 shows that if the model built by firm A is not differentiated, then no matter their price, firm B can always set a price that limits firm A's revenue. The implication is that the only way for firm A to profit is to either improve their model to specialize on at least one task, or use external factors (e.g., marketing).
>
>
> ---
>
> > W2) The main technical contribution to me is the modeling the GenAI’s prompt-based service. The analysis seems very standard.
>
>
> We agree that the key methodological contribution is the modeling and problem formulation of the pricing problem in generative AI. The goal of our follow-through analysis is to derive insights, rather than new algorithms. Further, we believe that our model is the first step towards more advanced analyses of generative AI pricing.
>
> ---
>
> > W3) I am not fully convinced that in the current GenAI market, the companies are following the dynamics of model A and model B in the paper. All the companies should be keeping updated their accuracy and be almost homogenous in terms of pricing.
>
> `Related: U4W8-Q1`
>
> This is a great point that our analysis also agrees with and we will include the following discussion in our revision.
>
> Our analysis in Proposition 1 shows that firms should frequently introduce new, better models, in addition to competing on price. Specifically, if the model performance is weak (i.e., $\kappa_2 / \kappa_1$ is large), then it doesn't matter what price they set because their competitor can set a more competitive price. Thus, we must continue to improve our models -- but by how much? Proposition 1 states that to guarantee revenue, we should ensure the model sufficiently outperforms competitors on at least one task (relative to other tasks), which allows the product to be sufficiently differentiated. This may be more efficient than competing on every task.

---

> > ### Comment · Reviewer_CY6H · 2024-08-11
> >
> > I have read the rebuttal carefully. Thanks for your efforts and your responses.

---

> > > ### Author Response · Authors · 2024-08-14
> > > **Thanks**
> > >
> > > Thank you very much for your valuable feedback and comments on this work.

---

### Official Review · Reviewer_aKR2 · 2024-07-13

**Soundness:** 3
**Presentation:** 3
**Contribution:** 2
**Rating:** 6
**Confidence:** 3

**Summary:**

* The paper explores the optimal pricing problem for generative AI services in a two-firm Stackelberg competition setting.
* In the setting, two generative AI firms compete over users. Each generative model is characterized by a fixed price $p$ per query, and success probabilities $(V_1, \\ldots, V_T)\\in[0,1]^T$ for each task. The demand for task $t$ at price $p$ is characterized by a non-increasing demand function $D_t(p)$. Users are assumed to query the generative model repeatedly until a satisfactory result is obtained, and success probability of each query ($V_t$) is assumed to be independent and identical. Two generative AI firms compete over users in a Stackelberg setting (firm A sets prices first, firm B follows), and their goal is to maximize revenue.
* For firm B, Theorem 1 characterizes the price optimization problem as a piecewise optimization problem. For firm A, Theorem 2 characterizes the price optimization problem as a bi-level optimization problem. Section 5 assumes an exponential parametric form for the demand function, and derives a globally-optimal solution based on them.

**Strengths:**

* Topic of the paper is well-motivated.
* Paper is well-written. Presentation is clear and very easy to follow.
* Assumptions are introduced gradually and explicitly, and seem to be justified by existing literature.
* The paper proposes guidelines for pricing in practical scenarios.

**Weaknesses:**

* The analysis seems to rely on the assumption that a full taxonomy of tasks is available, and that demand curves for each task are independent. However, due to the general-purpose nature of generative models, tasks may change over time, and therefore identifying and estimating all demand curves may be infeasible.
* Assumed cost model (pay-per-query) doesn’t seem to be common - Current common pricing schemes for generative models are per-token (e.g, OpenAI’s GPT-4 API), or per-month (e.g, ChatGPT), possibly inducing different incentive structures.
* Code is not provided, making it harder to reproduce results and build upon them.

**Questions:**

* What is the computational complexity of the optimization problem in Theorem 1?
* In the presented setting, can the firms benefit from colluding?
* What are the expected consequences of having more than two firms participating in the market?
* How would results change if the generative model was assumed to be able to "fail" in generating a satisfactory response? (either due to lack of ability, or users that churn before reaching satisfactory results)
* How would results change if the two firms are allowed play simultaneously?
* Minor typos:
  * L223 - “Theorem holds..” - Missing theorem number?
  * L239 - “cn” - Can?
  * L356 - “assumes” - assume?
  * L357 - “fims” - firms?

**Limitations:**

Most limitations are discussed explicitly, and in sufficient detail. I feel that the paper can benefit from discussing the additional limitations that appear above.

---

> ### Author Rebuttal · Authors · 2024-08-04
>
> Thank you for your positive comments on our motivation, clarity of writing, and overall presentation of our work.
>
> ---
> > W1) The analysis seems to rely on the assumption that a full taxonomy of tasks is available, and that demand curves for each task are independent. However, due to the general-purpose nature of generative models, tasks may change over time, and therefore identifying and estimating all demand curves may be infeasible.
>
> `Related: 6DrW-Q2.`
>
> We agree that for general LLM consumers, the number and type of tasks will vary over time. In practice, a company can update prices to reflect changing user demands. Our most effective use-case is for consistent/stationary users, such as companies that are building tools ontop of LLM APIs (e.g., PDF AI Chat software). These users have relatively consistent tasks and can quantify task statistics, user demand, and success rates.
>
> ---
> > W2) Assumed cost model (pay-per-query) doesn’t seem to be common - Current common pricing schemes for generative models are per-token (e.g, OpenAI’s GPT-4 API), or per-month (e.g, ChatGPT), possibly inducing different incentive structures.
>
> `Related: USe1-Q2.`
>
> We agree that most payment structures are per-token or subscription-based. Our per-prompt pricing is **a stylistic choice to reduce notation and our results automatically extend to pricing per-token** with a small change of variables. We will update our paper with the below extension.
>
> With per-token pricing, each task $t$ has a different average price $p_t = \theta_t p_0$ where $p_0$ is the price-per-token and $\theta_t$ is the average prompt length for the task (equivalently $q_t = \phi_t q_0$). Here, $p_0$ and $q_0$ are the price variables to optimize, whereas $\theta_t$ and $\phi_t$ are fixed parameters.
>
> For task $t$, a user will prefer model B if $\theta_t \frac{p_0}{V_t} \leq \phi_t \frac{q_0}{W_t}$ (i.e., Eqn 1). Using this principle, the corresponding pricing optimization problems (Eqn 2) can be solved by  redefining the competitive ratio between models on each task to $\kappa_t := \frac{\phi_t V_t }{\theta_t W_t}$. This redefined $\kappa_t$ can be substituted into the subsequent analyses to recover all our theoretical results.
>
> We agree that subscription pricing is an important setting, but it is typically for more general consumers. We will explore this problem in future work.
>
> ---
> > Q1) What is the computational complexity of the optimization problem in Theorem 1?
>
> The problem is a max of $T$ single-variable inner optimization problems with only bounding constraints. If the demand functions $D_t(p)$ are differentiable, each inner problem can be solved by gradient descent. For standard choices of $D_t(p)$ (e.g., exponential, linear), we can directly check the zero-derivative and boundary points. For example under Assumption 2, the problem automatically reduces to checking $2T$ possible values of $p$.
>
> ---
> > Q2) In the presented setting, can the firms benefit from colluding?
>
> Thanks for this great question! Our analysis reveals that for a company to maximize revenue, they should be particularly specialized in at least one task (Proposition 1). This is the generative AI equivalent of product differentiation in  goods. Intuitively, two companies could collude by specializing their models to different tasks (e.g., one company focuses on programming and quantitative tasks, while another focuses on language and writing tasks), thereby ensuring both companies receive revenue. However, quantifying this effect, as well as the social outcomes, is non-trivial and requires extensive effort. We would be happy to explore this more thoroughly in future work.
>
> ---
> > Q3) What are the expected consequences of having more than two firms participating in the market?
>
> `Related: U4W8-W1, 6DrW-Q5.`
>
> We will include this point in our updated paper. Our pricing problems can be extended to multiple competitors by taking the highest price-performance ratio for each task. If a company $(p, V_t)$ has two competitors, $(q, W_t)$ and $(r, X_t)$, a user will prefer the first model over all others if  $p/V_t \leq \max( q/W_t , r/X_t)$. This revised version of Eqn 1 can be used to re-define new pricing problems and recover Theorem 1 & 3 by re-defining $\kappa_t$ as a worst-case competitive ratio.
>
> ---
> > Q4) How would results change if the generative model was assumed to be able to "fail" in generating a satisfactory response? (either due to lack of ability, or users that churn before reaching satisfactory results)
>
> `Related: USe1-Q1.`
>
> This is a great question that we will include in our updated paper! **Our framework generalizes to the case where for each task $t$, the user churns after a maximum $T_t$ rounds.** Here, the total number of rounds for a task is a Truncated Geometric distribution (`omitted details for space, see USe1-Q1`). Using the expected value of this distribution, we can re-derive Eqn 1, revise the pricing problem (Eqn 2), and redefine the competitive ratio $\kappa_t$ between models to recover all of our theoretical results. Our main insights stay the same.
>
> ---
> > Q5) How would results change if the two firms are allowed play simultaneously?
>
> `Related: 6DrW-Q1.`
>
> Mathematically, we could explore a simultaneous competition framework where neither firm knows the price that their opponent will set a priori. Here, the pricing solution in Theorem 1 is not possible. A potential strategy is to use the robust problem in Theorem 2, which if solvable, would guarantee a non-zero revenue. However, this strategy is not optimal and may not achieve a Nash equilibrium if two companies are simultaneously playing.
>
> We agree that the simultaneous pricing problem is important, but it requires non-trivial analysis to determine equilibrium-achieving conditions. Since the sequential setting characterizes recent events where ChatGPT held a first-mover position over competitors, we focus on the sequential problem. We plan to explore the simultaneous problem in future work.

---

> > ### Comment · Reviewer_aKR2 · 2024-08-14
> >
> > Thank you for the thorough response, and for the helpful clarifications! I have no further questions.

---

> > > ### Author Response · Authors · 2024-08-14
> > > **Thanks**
> > >
> > > Thank you very much for your valuable feedback and comments on this work.

---

### Official Review · Reviewer_6DrW · 2024-07-13

**Soundness:** 2
**Presentation:** 3
**Contribution:** 2
**Rating:** 5
**Confidence:** 4

**Summary:**

This paper studies the pricing and competition of companies providing services using generative AI models. This paper proposes a stylized economic model that abstracts away from the technical details, and in particular, considers two companies entering the market sequentially. This paper assumes that the customer chooses the model that is cheaper per prompt, and each company solves a revenue maximization problem. Based on the model and analysis, this paper argues that companies should adopt different pricing strategies based on their order of entering the market.

**Strengths:**

1. The pricing of generative models is an interesting and timely matter

**Weaknesses:**

1. Several fundamental model setups and assumptions are not very justified.
1. The insights derived are not very informative nor it provide very meaningful guidance.  Overall, I understand that the parsimonious model aims to highlight certain main trade-offs, but this paper somewhat errs on the side of oversimplification.
1. Another big concern is that: the nature of analysis in this paper is very stylized and economic, and it is unclear if this is the right venue for such kind of papers..

**Questions:**

1. Is the sequential order of companies true? Would simultaneous competition be more reasonable? Furthermore, if so, do they get any first-mover advantage? The current model assumes that the first company is disadvantaged due to a lack of price information.
1. The current modeling seems to assume that customers are very rational and could make a sensible decision based on knowledge of price and number of prompts $n$. What about the cases when consumers do not have a good idea of the number of prompts needed or the success rate? What if customers have heterogeneous types or they have heterogeneous tasks?
1. The i.i.d. assumption of $V_t$ is concerning. Due to the conversational nature, the prompts are naturally highly correlated, and thus the independence of the success rate is not justified.
1. Can authors start the motivation with some more practical background?
1. To what extent does the current analysis extend to a more general scenario with multiple companies?
1. Punctuations are missing from several math equations.

---

> ### Author Rebuttal · Authors · 2024-08-04
>
> Thank you for your positive comments on the interest and timeliness of our work.
>
> ---
>
> > Q1) Is the sequential order of companies true? Would simultaneous competition be more reasonable? Furthermore, if so, do they get any first-mover advantage? The current model assumes that the first company is disadvantaged due to a lack of price information.
>
> `Related: aKR2-Q5.`
>
> Our sequential order of companies is motivated by real-world events of how ChatGPT had a large first-mover advantage before competing products were released. Furthermore, competitors were able to set their prices after observing the market performance of ChatGPT. Even now, firms release different models with prices asynchronously.
>
> Mathematically, we could explore a simultaneous competition framework where neither firm knows the price that their opponent will set. Here, Theorem 1 is not possible since it requires competitor prices. A potential strategy is to use the robust problem of Theorem 2. However, this may not achieve a Nash equilibrium if both companies simultaneously use it.
>
> We believe that the simultaneous pricing problem is important, but it requires non-trivial analysis to determine equilibrium-achieving conditions. We focus on the sequential problem in this work due to how it captures the real-world events, but we plan to study the simultaneous problem in future work.
>
>
> ---
>
> > Q2) The current modeling seems to assume that customers are very rational and could make a sensible decision based on knowledge of price and number of prompts $n$. What about the cases when consumers do not have a good idea of the number of prompts needed or the success rate? What if customers have heterogeneous types or they have heterogeneous tasks?
>
> `Related: aKR2-W1.`
>
> Our ssumption that users are rational and know the expected number of prompts is reasonable for large-scale, commercial users of LLM applications. For example, the user could be a company building software ontop of LLM API calls (e.g., PDF AI Chat tools). They would be informed on task statistics and success rate.
>
> Further, while our stylized model is defined for a 'single user with different tasks', the techniques naturally extend to multiple heterogeous users by aggregating the pricing objective (Eqn 2) into an expectation over the distribution of different types of users. Moreover, we expect that as a first-mover disadvantage exists for one user, an analogous disadvantage should exist over general multiple heterogeneous users.
>
> We agree that consumers who do not know their task statistics or success rates, may not be rational, but may be subject to additional factors, such as marketing or word-of-mouth adoption. This is an interesting extension that we will discuss in our paper and study in future work.
>
> ---
>
> > Q3) The i.i.d. assumption of $V_t$ is concerning. Due to the conversational nature, the prompts are naturally highly correlated, and thus the independence of the success rate is not justified.
>
>
> This is a great point. We agree that for some tasks, the success probability and number of rounds may not be independent. **Our framework and all results extend to arbitrary distributions of the number of prompting rounds required to complete a task $n(V)$.** Our initial Geometric assumption was a stylistic choice to make the downstream insights more interpretable, but we will revise our paper with the below generalization.
>
>
> For any distribution for $n(V)$ with finite expected value $E[n(V)]$, the principle for determining whether company B $(p, V)$ is preferred over company A $(q, W)$, i.e., Eqn 1, is $p E[n(V)] \leq q E[n(W)]$ and the pricing problem is
> $R(p|q) := p \sum_t D_t(p)\mathbf{1} [ p E[n(V_t)] \leq q E[n(W_t)] ]$. This optimization problem can still be solved using Theorem 1 if we re-define the competitive ratio between tasks as $\kappa_t := \frac{E[n(W_t)]}{E[n(V_t)]}$. Using this $\kappa_t$, we can recover all of our theoretical results.
>
>
> ---
>
> > Q4) Can authors start the motivation with some more practical background?
>
>
> We will revise our paper introduction to better highlight our motivation.
>
> One motivation of our work was the first-mover position of ChatGPT, which led to several competing products being released shortly after. This technology invited a new paradigm of interacting with a single model to solve various different tasks via multiple prompts. While this product has seemed to be dominant, there are several competing products that all perform relatively similarly on these tasks [1]. For this emerging market, when should users prefer one competing model over another, and what is the market advantage in the current rapid pace of development where companies are quickly trying to release better models as fast as possible? We study these problems by analyzing the relationships between model performance, competition, and price.
>
>
> [1] https://scale.com/leaderboard
>
>
> ---
>
> > Q5) To what extent does the current analysis extend to a more general scenario with multiple companies?
>
> `Related: U4W8-W1, aKR2-Q3.`
>
> Our duopoly framework and corresponding pricing problems (Eqn 2) naturally extend multiple companies by simply taking the highest price-performance ratio for each task. If company A $(p, V)$ has two competitors,  $(q, W)$ and $(r, X)$, a user will prefer company A over the competitors if $\frac{p}{V_t} \leq \max( \frac{q}{W_t}, \frac{r}{X_t}  )$ (i.e. Eqn 1). We can then rewrite the pricing problem (Eqn 2), re-define the competitive ratio $\kappa_t$ to be a worst-case ratio over all competitors, and directly recover many of the theoretical results (e.g., Theorem 1, Theorem 3) in our paper.

---

> > ### Comment · Reviewer_6DrW · 2024-08-07
> > **Score adjusted**
> >
> > I read the rebuttals thoroughly and appreciate the authors' detailed and thoughtful responses to my questions as well as similar concerns by other reviewers. I have thus adjusted my score accordingly.

---

> > > ### Author Response · Authors · 2024-08-08
> > > **Thanks**
> > >
> > > Thank you very much for the improved score. We appreciate your feedback and the overall review process. We are happy to continue discussing the paper and answering any further questions or concerns that come up.

---

### Official Review · Reviewer_U4W8 · 2024-07-13

**Soundness:** 3
**Presentation:** 3
**Contribution:** 2
**Rating:** 5
**Confidence:** 4

**Summary:**

The paper identifies some unique characteristics of modern generative AI software which affect their pricing. It uses a notion of user cost-effectiveness to compare two models, capturing the cost per prompt and the number of prompting rounds needed to reach a satisfactory answer. The authors propose to model the pricing problem as a game between two companies sequentially releasing their models before users choose their preferred model for each task. A set of tasks is assumed and each task is priced separately. They show that the price optimization problem is piecewise continuous i.e. the companies must choose a subset of tasks on which to be cost-effective and forgo revenue on other tasks. Their analysis indicates that being the first to market may become cost-ineffective, as companies entering the market later can benefit from knowing their competitor's pricing.

**Strengths:**

1. Pricing digital goods and services is an interesting problem in general due to their unique characteristics. The paper is a nice early attempt to characterize and price generative AI.

2. The modeling is simple and accounts for user satisfaction, demand, and accuracy of the model for different tasks.

3. Analysis with the above model reveals to the player (company) that companies have to choose a subset of tasks on which they can be cost-effective and forgo revenue on other tasks. It also shows that when the tasks are sufficiently similar, then the first-to-market may become cost-ineffective on all tasks regardless of the pricing.

**Weaknesses:**

1. The setup seems far from reality and loosely related to generative AI. First, in real life even if we assume two companies, there is nothing prohibiting them from updating their prices any number of times and the first mover is probably always in a better position due to customer acquisition, branding, and acquiring real data, system testing, etc. So the analysis seems counterintuitive.


2. The connection with generative AI is not very clear. What unique characteristics of generative AI are actually used in the pricing model? It is generally applicable to any model with different performance on a set of tasks.


3. There are no real data experiments or simulations provided in support of the modeling assumptions and analytical results. It is understandable the reality is complex and there are a lot of variables involved in pricing. It would be unreasonable to expect to model all of that but I do expect this study to be a bit more closer to reality than it is. It could be done by showing the real data and formulating the modeling assumptions from it and the pricing game with multiple parties going over multiple rounds and also improving their models. It would be great if with the proposed model the future price can be predicted accurately.

**Questions:**

See above.

**Limitations:**

Yes.

---

> ### Author Rebuttal · Authors · 2024-08-04
>
> Thank you for your positive comments on the uniqueness of our problem as well as the simplicity and comprehensiveness of our model.
>
> ---
>
> > W1) The setup seems far from reality and loosely related to generative AI. First, in real life even if we assume two companies, there is nothing prohibiting them from updating their prices any number of times and the first mover is probably always in a better position due to customer acquisition, branding, and acquiring real data, system testing, etc. So the analysis seems counterintuitive.
>
>
> Thank you for several important points. Below, we justify our formulation to show that it reasonably captures real-world trends and we will revise our paper with this discussion.
>
>
>
> 1. ( `Related: 6DrW-Q5, aKR2-Q3.`) Our framework can be extended to multiple companies by taking the highest price-performance ratio for each task. If a company $(p, V_t)$ has two competitors with $(q, W_t)$ and $(r, X_t)$, then a user will prefer the first model if  $\frac{p}{V_t} < \max( \frac{q}{W_t} , \frac{r}{X_t})$ (i.e., Eqn 1). With this, we can revise the pricing optimization problems (Eqn 2) and automatically extend Theorem 1 & 3 with a re-defined competitive ratio $\kappa_t$. Although the general first-mover's problem (Theorem 2) becomes more difficult to solve, note that facing multiple competitors should preserve a first-mover disadvantage effect similar to what we currently observe.
>
>
>
> 2. (`Related: CY6H-W3.`) We agree that real-world pricing may be dynamic with firms updating their prices over time. Our model is a natural baseline for this setting, since each turn of a dynamic problem can be naively tackled using static decisions. Furthermore, our results reveal key insights about the dynamic setting. Proposition 1 shows that if model performance is not competitive (i.e., $\kappa_2 / \kappa_1$ is large), then for any price that the company sets, their competitor can always beat them. **Thus, companies cannot just update prices, but must also continually improve their models or else be eventually beaten.**
>
>
>    We believe a full dynamic pricing study of this problem must include both pricing and model performance as variable, making it a difficult open problem. Our work is the necessary first step to tackling the extension.
>
>
>
> 3. We agree that factors such as customer acquisition, branding, and the AI data/testing flywheel also significantly impact revenue. However, modeling every such component introduces complexity without meaningfully changing our key takeaway.
>
>    Specifically, we show that the first-mover advantage is most meaningful when the model has a particular specialization to at least one task (Proposition 1). This is a quantifiable example of the product differentiation principle and is relevant especially now as the top models perform statistically similarly on most application areas [1]. We observe the importance of pricing and product differentiation in, for example, releases of differentiated models (e.g., Claude 3 Opus, Sonnnet, Haiku), and price drops (e.g., GPT3.5 in Nov 2023 and Jan 2024).
>
>
>
> [1] https://scale.com/leaderboard
>
>
> ---
>
> > W2) The connection with generative AI is not very clear. What unique characteristics of generative AI are actually used in the pricing model? It is generally applicable to any model with different performance on a set of tasks.
>
> Our revision will clarify this below point. Our framework is suitable for modern LLMs (and VLMs/Multimodal LLMs) via two key traits: (1) there are multiple types of tasks; (2) users interact via prompting over multiple rounds. The combination of these two traits is not found in older ML technology. They imply that there is a single price variable affecting all tasks (i.e., price-per-token/price-per-prompt), and that users seek to minimize the number of prompting rounds. The prompting trait especially differentiates the use-case from general multi-task models, where a user can only input a task instance once.
>
>
> ---
>
> > W3) There are no real data experiments or simulations provided in support of the modeling assumptions and analytical results. It is understandable the reality is complex and there are a lot of variables involved in pricing. It would be unreasonable to expect to model all of that but I do expect this study to be a bit more closer to reality than it is. It could be done by showing the real data and formulating the modeling assumptions from it and the pricing game with multiple parties going over multiple rounds and also improving their models. It would be great if with the proposed model the future price can be predicted accurately.
>
>
> To the best of our ability, our model assumptions are based on real data and justification from the literature (as positively remarked by `aKR2`). For example, we studied user interactions from Chatbot Arena to validate that users will often interact for multiple rounds until they are satisfied.
>
>
> Unfortunately, numerically predicting future price accurately requires access to the true demand functions for LLMs, which to our knowledge, is not publicly available. Thus, our numerical analysis is limited to synthetic scenarios (e.g., Fig. 2, Fig. 3) where we impose a hypothetical demand function and demonstrate the outcome.
>
>
> We also agree that observing multiple parties over multiple rounds to be a very interesting problem, but this changes the problem to a dynamic pricing setting, which requires more theoretical analysis. We envision exploring the dynamic problem in future work both theoretically and with synthetic numerical experiments.

---

> ### Comment · Reviewer_U4W8 · 2024-08-08
>
> Thank you for the rebuttal. I have read it and my concerns about the connection with generative AI and "realisticness" of the setup/assumption remain the same. If we just change  $p$  to price per inference call, then I believe the whole story can be written for any other machine learning model.
>
> The paper does not do justice to the title/motivation. It is overall an oversimplified economic setup leading to some insights which seem unrealistic and of little value that are tied with the generative AI hype. I expected to learn more from this paper, particularly what makes the pricing problem interesting and challenging and some realistic solutions even in the 2 player setting.
>
> The value of data is largely ignored. I believe a first mover has a significant advantage in obtaining the real data and improve their product and this cycle of rich gets richer continues i.e. better model --> more users --> more data --> better model.
>
> Having said that, I do not expect a research paper will solve the problem in entirity, but I hope to learn more about the problem and why the solution makes sense. I suggest taking a look at the following references, ( some of them are non-academic blog posts ).
>
> 1. https://every.to/p/how-to-price-generative-ai-products
>
> 2. https://sada.com/blog/generative-ai-pricing/
>
> Pricing data and models
>
> 3. https://dl.acm.org/doi/abs/10.1145/3328526.3329589
>
> 4. https://arxiv.org/pdf/1206.6443
>
> 5. https://openreview.net/pdf?id=Y6IGTNMdLT
>
> 6. https://arxiv.org/abs/2312.04740
>
> 7. https://arxiv.org/pdf/2108.07915
>
> Based on the current state, I reserve my borderline score.

---

> ### Author Response · Authors · 2024-08-09
> **Thanks for the feedback and additional references**
>
> Thanks for the detailed feedback and discussion. The related research literature on data pricing and fair model valuation is relevant and we are happy to revise our paper with this discussion.
>
> ---
>
> > If we just change  to price per inference call, then I believe the whole story can be written for any other machine learning model.
>
> Pricing per-prompt/per-token in generative AI and pricing per-instance in classical ML models are different because generative models uniquely permit multiple calls to obtain a good answer. For example in a reading QA task, if a promptable LLM gets a wrong output, we query again with a revised prompt, thus paying twice to get the right answer; if a classical ML model is incorrect, the user has no such recourse and must deal with the incorrect answer. This difference necessitates different model valuation strategies (for instance, [1] uses $V\times$ the amount a user is willing to pay for a marginal performance increase). **We believe that the user valuation of a generative AI model should be $p E[n(V)]$**. Note this valuation would not make sense for classical ML products.
>
> Our valuation and corresponding Eqn 1 specializes all our downstream results, whereas classical ML models would require an alternative to Eqn 1 (e.g., [1]), which then would require a different revenue function and optimal solution structure. Most importantly, our valuation function naturally combines with the property of multiple tasks with different demands, thereby giving our differentiation property (Proposition 2), which does not have any place in classical ML products.
>
> [1] https://dl.acm.org/doi/abs/10.1145/3328526.3329589
>
> ---
>
> > The value of data is largely ignored. I believe a first mover has a significant advantage in obtaining the real data and improve their product and this cycle of rich gets richer continues i.e. better model --> more users --> more data --> better model.
>
> We agree that the value of data is an important factor. We characterize data into two types, internal and external, that have different properties. We will revise our paper with the following discussion.
>
> 1. **Data that the model developer obtains independently:** This data directly impacts model performance, e.g., via a scaling law $V \propto a n^{-b}$ where $n$ is the amount of data. In our pricing problem, **the effect of this data can be directly incorporated by substituting the scaling law into the model performance parameter.** To this end, all our results that speak to requiring a minimium model performance also imply a minimum amount of training data required, e.g., Proposition 2 imposes a maximum value on $\kappa_2 / \kappa_1 := \frac{V_2 W_1}{V_1 W_2}$ as functions of $n$).
>
> 2. **Data collected via the AI flywheel:** We agree that this is important for model development, especially in the long-tail. We also agree that your suggestion is the natural next step for our work. However, this analysis presents the following concerns:
>
>    - *Task-specific data may not always be available from the flywheel.* For example, ChatGPT offers data collection opt-out for customers and does not train on enterprise data [2]. Thus, the relationship between the AI flywheel and model performance on custom tasks is indirect (e.g., releasing a better model early to draw more users may improve overall/general model quality [3], but have limited effect on some specific tasks).
>
>    - *Analyzing this requires a more complex framework for which the current work is a necessary precursor.* For example, the extension requires a dynamic problem with time-variant variables for price and performance (or equivalently, data). This complexity, combined with the length and novelty of our current study, motivates us to leave the dynamic extension for follow-up work.
>
> [2] https://openai.com/enterprise-privacy/
>
> [3] Huseyin Gurkan and Francis de Véricourt. Contracting, pricing, and data collection under the ai flywheel effect.

---

> > ### Comment · Reviewer_U4W8 · 2024-08-11
> >
> > Thank you for the response. I appreciate the paper as an early attempt towards pricing in generative AI. However, I am not convinced about the main technical challenges/distinctions specific to pricing in this context. Thus I will keep my original recommendation of borderline accept. I'd also encourage authors to include a more comprehensive related work and a clear delineation of the technical challenges specific to this setting in the next version. Thanks again and good luck!

---

### Official Review · Reviewer_USe1 · 2024-07-16

**Soundness:** 3
**Presentation:** 4
**Contribution:** 4
**Rating:** 8
**Confidence:** 3

**Summary:**

This paper develops a theoretical model of users deciding which generative AI system to use based on the price and probability of performing a task satisfactorily. Based on this theoretical model, the paper then explains how a firm should price their generative AI model in response to other firms, and then how a firm should price their model knowing that other firms will respond. They show that, depending on the ratio of ability to perform a task between the two models, a firm will either get no revenue since they are not price-competitive for anything, charge low enough prices to be price competitive on some tasks, or be so much better at a task than the other firm that they can charge prices which are bound based only on decreasing demand as price increases.

**Strengths:**

Significance: This paper builds a model for analyzing pricing for generative AI services which can be used for multiple different tasks, which is a question which is very broadly applicable to analyzing AI pricing.
Originality: The pricing model is original, with several new definitions and problem formulations.
Quality: The paper has proofs supporting its major claims.
Clarity: The paper is very clearly written, and I found its figures to be exceptionally helpful in understanding the claims of the paper.

**Weaknesses:**

1) The generative AI task performance model seems somewhat unrealistic -- it assumes that given enough user queries the model will always eventually succeed.

**Questions:**

1) Is there an alternative way of modeling task performance that does not assume that the AI system will always succeed?
2) Different tasks might use different numbers of tokens while performing the task, which would result in higher costs for that specific task. Would it change the results a lot if all tasks shared a per-token price but had variable costs for the different tasks based on how many tokens the task requires?

**Limitations:**

The authors have adequately addressed the limitations and potential negative societal impact for their work.

---

> ### Author Rebuttal · Authors · 2024-08-04
>
> Thank you for your clear summary of our work, your positive comments on the originality and significance/broad applicability of our work, as well as on the clarity of our writing.
>
> ---
>
> > Q1) Is there an alternative way of modeling task performance that does not assume that the AI system will always succeed?
>
> `Related: aKR2-Q4.`
>
> This is a great point that we will update in our paper. We agree that for some complex tasks $t$, a user may give up after some maximum number of rounds $T_t$ before the model succeeds. **Our results all extend to this scenario (or any finite-mean distribution) with a small change of variables.**
>
>
> In a setup where users terminate after some fixed $T_t$ rounds, the total number of rounds $n$ that a user prompts is a Truncated Geometric distribution
>
> $$Pr(n) =
> \begin{cases}
>     (1-V_t)^{n-1}~V_t & \text{ for } 1 < n < T_t \\\\
>      (1-V_t)^{T_t-1}  & \text{ for } n = T_t
> \end{cases}
> $$
>
> Using the expected value of this distribution, the corresponding Eqn 1, i.e., when user prefers model B $(p, V_t)$ over model A $(q, W_t)$, is $p( \frac{1-(1-V_t)^{T_t-1}}{V_t} + (1-V_t)^{T_t-1} ) \leq q( \frac{1-(1-W_t)^{T_t-1}}{W_t} + (1-W_t)^{T_t-1} )$. This principle plugs into a revised pricing problem (Eqn 2), and requires a re-defined competitive ratio between two models on a task:
>
> $$
> \kappa_t := \frac{ \frac{1-(1-V_t)^{T_t-1}}{V_t} + (1-V_t)^{T_t-1} }{ \frac{1-(1-W_t)^{T_t-1}}{W_t} + (1-W_t)^{T_t-1} }.
> $$
>
> With this re-defined $\kappa_t$, we can solve the pricing problem using Theorem 1 and recover all theoretical discussion and insights in our paper. Our main conclusions stay the same.
>
>
> The maximum number of rounds $T_t$ is a parameter that can be estimated from historical data. For example in the Chatbot Arena dataset, users spend on average 1.3 rounds before determining the generative model output as satisfactory (i.e., $T = 2$ for many tasks).
>
> ---
>
> > Q2) Different tasks might use different numbers of tokens while performing the task, which would result in higher costs for that specific task. Would it change the results a lot if all tasks shared a per-token price but had variable costs for the different tasks based on how many tokens the task requires?
>
> `Related: aKR2-W2.`
>
> We agree that different tasks use different number of tokens, and thus, have different costs. **This extension will recover all our original results.** We will include this in revision.
>
>
> To account for tasks with different prompt types, we define the price-per-prompt for each task $t$ as $p_t = \theta_t p_0$ where $p_0$ is the price-per-token and $\theta_t$ is the average number of tokens-per-prompt for task $t$ (equivalently $q_t = \phi_t q_0$ for model A). Note that $p_0$ and $q_0$ are the price variables that the company can set whereas $\theta_t$ and $\phi_t$ are fixed parameters. Following the same steps as before, we can revise Eqn 1 to $\theta_t \frac{p_0}{V_t} \leq \phi_t \frac{q_0}{W_t}$ and also update the pricing problem Eqn 2. All our results automatically extend after we re-define the competitive ratio to $\kappa_t := \frac{\phi_t V_t}{\theta_t W_t}$.

---

> > ### Comment · Reviewer_USe1 · 2024-08-14
> >
> > Thank you for the detailed response. The responses increased my opinion of the paper both by adding new results and being simple enough that it increases my opinion of the original contributions. I am increasing my overall score from 6 to 8 and contribution score from 3 to 4.

---

> > > ### Author Response · Authors · 2024-08-14
> > > **Thanks**
> > >
> > > Thank you very much for the improved score and opinion on our paper. Thanks also for your valuable feedback and comments on this work.

---

### Author Rebuttal · Authors · 2024-08-04

We thank all the reviewers for their positive comments:
- Originality, novelty, and timeliness of our study (`USe1, U4W8, 6DrW, CY6H`)
- Broad applicability of our model & insightful analysis/guidelines/conclusions drawn (`USe1, U4W8, aKR2, CY6H`)
- Clarity of our written presentation (`USe1, aKR2, CY6H`)

Our work is motivated by the recent trends in generative AI (i.e., LLMs/Multimodal LLMs) where ChatGPT enjoyed a first-mover position. The prompt-based interaction combined with diverse set of applications make pricing these ML models different from others. We are not trying to prescribe exact prices, but instead we are studying the interaction between the performance vs price of these models when given competing alternative models.

Our general formulation allows us to derive high-level insights on the generative AI market. Our key insight is the observation that if model performance over different tasks fails to satisfy a relative ratio requirement (i.e., $\kappa_2 / \kappa_1$ is large), then the company's revenue can always be limited by a competing alternative.

We received questions mainly on several extensions of our model. With minor change of variables, **most extensions can be easily accommodated while preserving all our conclusions**. We summarize these extensions below and will include them in revision:

- **If users stop after a number of rounds or if prompts are non-i.i.d. (`USe1-Q1, 6DrW-Q1, aKR2-Q4`):** We can use any arbitrary distribution on the number of prompting rounds $n(V)$. In general, Eqn 1 becomes $p E[n(V_t)] \leq q E[n(W_t)]$ and the pricing problem (Eqn 2) changes to reflect this. If we re-define the competitive ratio to $\kappa_t := \frac{E[n(W_t)]}{E[n(V_t)]}$, then we can recover all our results.

- **Different prompt lengths and pricing per-token (`USe1-Q2, aKR2-W2`):** For task $t$, the price per-prompt becomes $p_t := p_0 \theta_t$ where $p_0$ is the price per-token and $\theta_t$ is the average prompt length (respectively $q_t := q_0 \phi_t$ for the competitor). Here, Eqn 1 becomes $\theta_t \frac{p_0}{V_t} \leq \phi_t \frac{q_0}{W_t}$ and the competitive ratio becomes $\kappa_t := \frac{\phi_t V_t}{\theta_t W_t}$. We can recover all our results.

- **If there are more than 2 companies (`U4W8-Q1, 6DrW-Q5, aKR2-Q3`):** If there are three models $(p, V_t), (q, W_t), (r, X_t)$, then users will prefer the first model if $p/V_t \leq \max(q/W_t , r/X_t)$.  We can update the pricing problem (Eqn 2) and still solve the latecomer's pricing problem (Theorem 1 & Theorem 3) by re-defining $\kappa_t := V_t / \max(W_t, X_t)$.

---

### Decision · Program_Chairs · 2024-09-25

**Decision:**

Accept (poster)

**Comment:**

The paper studies the problem of pricing generative AI technologies. Their setting assumes different tasks and two firms that sequentially price their tasks based on their performance to maximize their profit.

While the reviewers generally liked the motivation behind the paper and the analysis of the simplified model, they were concerned about several assumptions of the paper. These include the one-shot nature of the game, the assumption of iid results for repeated queries and the lack of any accounting for the cost it takes to generate the technology. The authors had provided to some of these concerns in the rebuttal. While I do not expect a published work to address all these issues, sI encourage the authors to incorporate these to the limitations/discussion of the paper. I would also encourage the authors to include a more in-depth discussion of why the game applies to generative AI companies more clearly in the final version.